# Large-scale cell-type-specific imaging of protein synthesis in a vertebrate brain

Or David Shahar, Erin Margaret Schuman*

Max Planck Institute for Brain Research, Frankfurt, Germany

**Abstract** Despite advances in methods to detect protein synthesis, it has not been possible to measure endogenous protein synthesis levels in vivo in an entire vertebrate brain. We developed a transgenic zebrafish line that allows for cell-type-specific labeling and imaging of nascent proteins in the entire animal. By replacing leucine with glycine in the zebrafish MetRS-binding pocket (MetRS-L270G), we enabled the cell-type-specific incorporation of the azide-bearing non-canonical-amino-acid azidonorleucine (ANL) during protein synthesis. Newly synthesized proteins were then labeled via 'click chemistry'. Using a Gal4-UAS-ELAV3 line to express MetRS-L270G in neurons, we measured protein synthesis intensities across the entire nervous system. We visualized endogenous protein synthesis and demonstrated that seizure-induced neural activity results in enhanced translation levels in neurons. This method allows for robust analysis of endogenous protein synthesis in a cell-type-specific manner, in vivo at single-cell resolution.

## Introduction

Protein synthesis is critical for remodeling synaptic proteomes, especially when this process is associated with information storage (*Sutton and Schuman, 2006*). Chemical stimuli and changes in behavioral states alter protein expression in the nervous system. It has been shown in different model organisms that protein synthesis, during or shortly after learning, is essential for the formation of long-term memory (*Davis and Squire, 1984*; *Agranoff et al., 1966*; *Agranoff and Klinger, 1964*; *Costa-Mattioli et al., 2009*). Despite the importance of neuronal protein synthesis for many biological processes such as learning (*Flexner et al., 1962*; *Hinz et al., 2013*; *Roberts et al., 2013*), stress responses (*Langebeck-Jensen et al., 2019*), and epilepsy (*Brooks-Kayal et al., 1998*; *Hinz et al., 2012*; *Baraban et al., 2005*; *Del Bel et al., 1998*), little is known about the endogenous neuronal protein synthesis levels and their changes in vivo.

Zebrafish are vertebrates that exhibit a variety of complex behaviors (*Orger and de Polavieja, 2017*) including swimming and rheotaxis (e.g. *Olszewski et al., 2012*; *Oteiza et al., 2017*; *Marques et al., 2018*), hunting (e.g. *Bianco et al., 2011*; *Semmelhack et al., 2014*), learning (eg. *Hinz et al., 2013*; *Roberts et al., 2013*; *Aizenberg and Schuman, 2011*; *Kenney et al., 2017*; *Ahrens et al., 2012*; *Valente et al., 2012*) and social behaviors (e.g. *Hinz et al., 2013*; *Peichel, 2004*; *Oliveira, 2013*; *Gerlai, 2014*; *Teles et al., 2016*; *Stednitz et al., 2018*; *Dreosti et al., 2015*). Moreover, a variety of neurological syndromes including epilepsy have been investigated (*Kundap et al., 2017*). The zebrafish larval brain is small and translucent, enabling high-resolution imaging of cells. The complexity of brain tissue, however, is still an issue. Neurons, in particular, have long processes, which are tightly entangled in their respective tissues. As such, monitoring protein synthesis levels with cell-type and temporal resolution has so far been impossible in zebrafish neurons, highlighting the need for the methodology developed here (*Figure 1A*).

Bio-orthogonal approaches based on metabolic precursors enable the labeling of nascent proteins or protein modifications (*Hinz et al., 2012*; *Beatty et al., 2006*; *Laughlin et al., 2008*), and have been combined with immunostaining to measure protein synthesis in specific cell types (*Liu and Cline, 2016*). These platforms have recently been coupled with genetic control,

*For correspondence:
erin.schuman@brain.mpg.de

Competing interests: The authors declare that no competing interests exist.

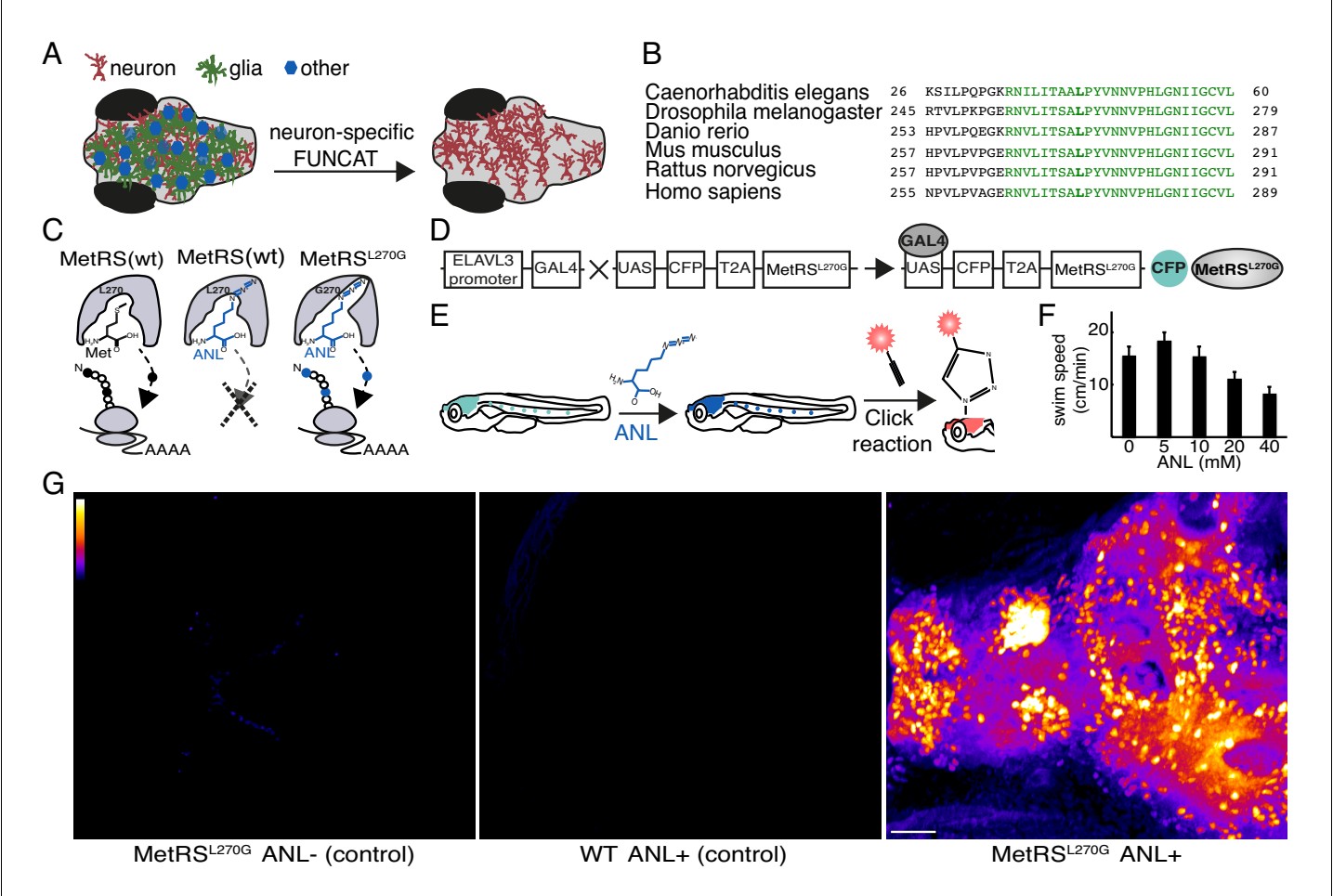

**Figure 1.** Cell-type-specific labeling of newly synthesized proteins. (**A**) Schematic demonstrating the method to visualize newly synthesized proteins in the brain in a cell-type-specific manner. (**B**) The protein sequence of the catalytic core domain of the MetRS in a number of species including *Danio Rerio* shows strong conservation (green). Leucine 270 (bold green) was mutated to Glycine to develop cell-type-specific metabolic labeling in zebrafish. (**C**) Schematic of the binding pocket of the MetRS and the ribosome during translation. The wt MetRS allows the charging of Met (black) that can be incorporated during translation initiation and elongation (left). The non-canonical amino acid ANL (blue), which contains an azide group, does not fit into the binding pocket of the wt MetRS, and thus is notincorporated into nascent protein (center). The mutant MetRS$^{L270G}$ can charge ANL, which is then incorporated into newly synthesized proteins (in cells expressing the MetRS$^{L270G}$). (**D**) Schematic of the UAS-CFP-MetRS$^{L270G}$ line transgene. Crossing the line with any Gal4-expressing line allows for the metabolic labeling of newly synthesized proteins in any accessible cell type. (**E**) A scheme demonstrating the use of the ELAVL3-Gal4:UAS-CFP-MetRS$^{L270G}$ line. Left: a zebrafish larva expressing the transgene in neurons (cyan). Following addition of ANL to the water bath, newly synthesized proteins in neurons incorporate ANL (blue). Right: a whole mount click reaction with a fluorescent alkyne reveals the newly synthesized proteins (red). (**F**) The effect of different ANL concentrations on swim speed after 24 hr of ANL exposure (measurement was done in the presence of ANL). 10 mM ANL, which had no significant effect on larvae swimming, was used in further experiments. N = 5 to 6 larvae for each concentration. (**G**) Projections of confocal images of zebrafish larval brains after click reactions demonstrating the specificity of fluorescently labeled nascent protein in the MetRS$^{L270G}$ larva treated with ANL, but not in controls. Scale bar = 50 μm.

The online version of this article includes the following video and figure supplement(s) for figure 1:

**Figure supplement 1.** ELAVL3-MetRS$^{L270G}$ Zebrafish larvae maintain light preference following exposure to ANL.

**Figure 1—video 1.** Shown are dorsal view confocal planes (top to bottom and back) of an ELAVL3-MetRS$^{L270G}$ zebrafish larva brain following 24 hr of ANL labeling followed by click chemistry with a fluorescent alkyne tag.

https://elifesciences.org/articles/50564#fig1video1

**Figure 1—video 2.** Shown are dorsal view confocal planes (top to bottom and back) of one ELAVL3-MetRS$^{L270G}$ zebrafish larva brain not incubated with ANL (ANL-) and two larvae incubated with ANL for 24 hr, one WT and one ELAVL3-MetRS$^{L270G}$.

https://elifesciences.org/articles/50564#fig1video2

allowing access to particular cell types. For example, the wild-type Methionyl-tRNA synthetase (MetRS) can be modified to enable the charging of a different azide-bearing non-canonical amino acid, the methionine analog azidonorleucine (ANL), which cannot be charged by the wild-type MetRS. By using cell-type-specific promoters, the mutant MetRS can be expressed in cell types of interest and the nascent proteins can be labeled via the administration of ANL, as has been demonstrated in *Caenorhabditis elegans* (*Yuet et al., 2015*), *Drosophila melanogaster* (*Erdmann et al., 2015*) and *Mus musculus* (*Alvarez-Castelao et al., 2017*).

To date, overall levels of protein synthesis within neurons across the entire intact brain have not yet been measured and imaged in any vertebrate. The existing protein synthesis reporters used for most single cell analyses rely on fluorescent tagging of individual protein species and therefore do not measure endogenous nascent protein levels. Here we demonstrate for the first time the ability to label and image in situ newly synthesized proteins in vivo in a cell-type-specific manner. We visualized nascent neuronal proteins across the entire animal. Combining a specific Gal4 reporter line with the UAS-MetRS$^{L270G}$ and adding ANL for various durations enabled cell-type-specific labeling with temporal control. We also demonstrate the sensitivity of the protein synthesis signal to alterations in neuronal activity.

## Results

### Cell-type-specific nascent protein tagging

To enable cell-type-specific non-canonical amino acid tagging in zebrafish, we cloned and mutated the zebrafish MetRS to introduce a point mutation (L270G) at a conserved position in the methionine binding pocket (*Figure 1B–C*). We developed a transgenic line in which the conditional expression of CFP and the MetRS$^{L270G}$ (separated by T2A) is under the control of the UAS enhancer. When these fish are crossed to a Gal4 line with a cell-type-specific promoter, cell-type-specific incorporation of azidonorleucine (ANL) into nascent proteins can be achieved (*Figure 1C–D*). In order to realize cell-type-specific labeling of newly synthesized proteins in neurons, we crossed the UAS-CFP-MetRS$^{L270G}$ line with a well-established pan neuronal driver Gal4 line, ELAVL3-Gal4 (*Figure 1D–E*). Following the addition of ANL, the charging of ANL by MetRS$^{L270G}$ and its incorporation into protein, a click reaction was performed and newly synthesized proteins were visualized (*Figure 1E*). In a previous study, we showed that zebrafish nascent proteins can be globally labeled with the non-canonical amino acid azidohomoalanine (AHA) via its addition to the swim water (*Hinz et al., 2012*). We added ANL to the swim water and determined whether there were any apparent behavioral effects. Freely swimming larvae were incubated with different concentrations of ANL for 24 hr (hrs) and the average swim speed was measured. We found that ANL concentrations of 10 mM or less did not affect larval swimming behavior (*Figure 1F*). Zebrafish larvae exhibit a preference for illuminated regions of their habitat (*Hinz et al., 2013*). We also observed that the ELAVL3-MetRS$^{L270G}$ larvae maintained their light preference in the presence of ANL (*Figure 1—figure supplement 1*). We next incubated larvae with 10 mM ANL for 24 hr, fixed the larvae and performed a whole-mount click reaction (see Materials and methods). Confocal images acquired across the brain revealed newly synthesized proteins in neurons in ELAVL3-MetRS$^{L270G}$ larvae that were incubated with ANL. Only weak fluorescence was detected in WT larvae that were incubated with ANL, or ELAVL3-MetRS$^{L270G}$ larvae that were not incubated with ANL (*Figure 1G*, for images of single z-sections see *Figure 1—video 1* and *Figure 1—video 2*).

### Detection of endogenous nascent proteins in different neuronal populations

We performed bio-orthogonal non-canonical amino-acid tagging (BONCAT) (*Dieterich et al., 2007*; *Dieterich et al., 2006*) on proteins extracted from WT or MetRS$^{L270G}$ larvae (4 dpf) after metabolic labeling (24 hr, 10 mM ANL) or MetRS$^{L270G}$ fish without metabolic labeling. Western blot analysis revealed an abundance of biotinylated nascent proteins, spanning various molecular weights, in head tissue from ANL-treated MetRS$^{L270G}$ larvae (*Figure 2A*, *Figure 2—figure supplement 1*) and only low background levels of biotinylated proteins in the controls (WT ANL+, MetRS$^{L270G}$ ANL- in *Figure 2A*). To examine the robustness and specificity of the labeling, we visualized newly synthesized proteins in different brain regions. We incubated 3 dpf larvae with 10 mM ANL for 24 hr, fixed

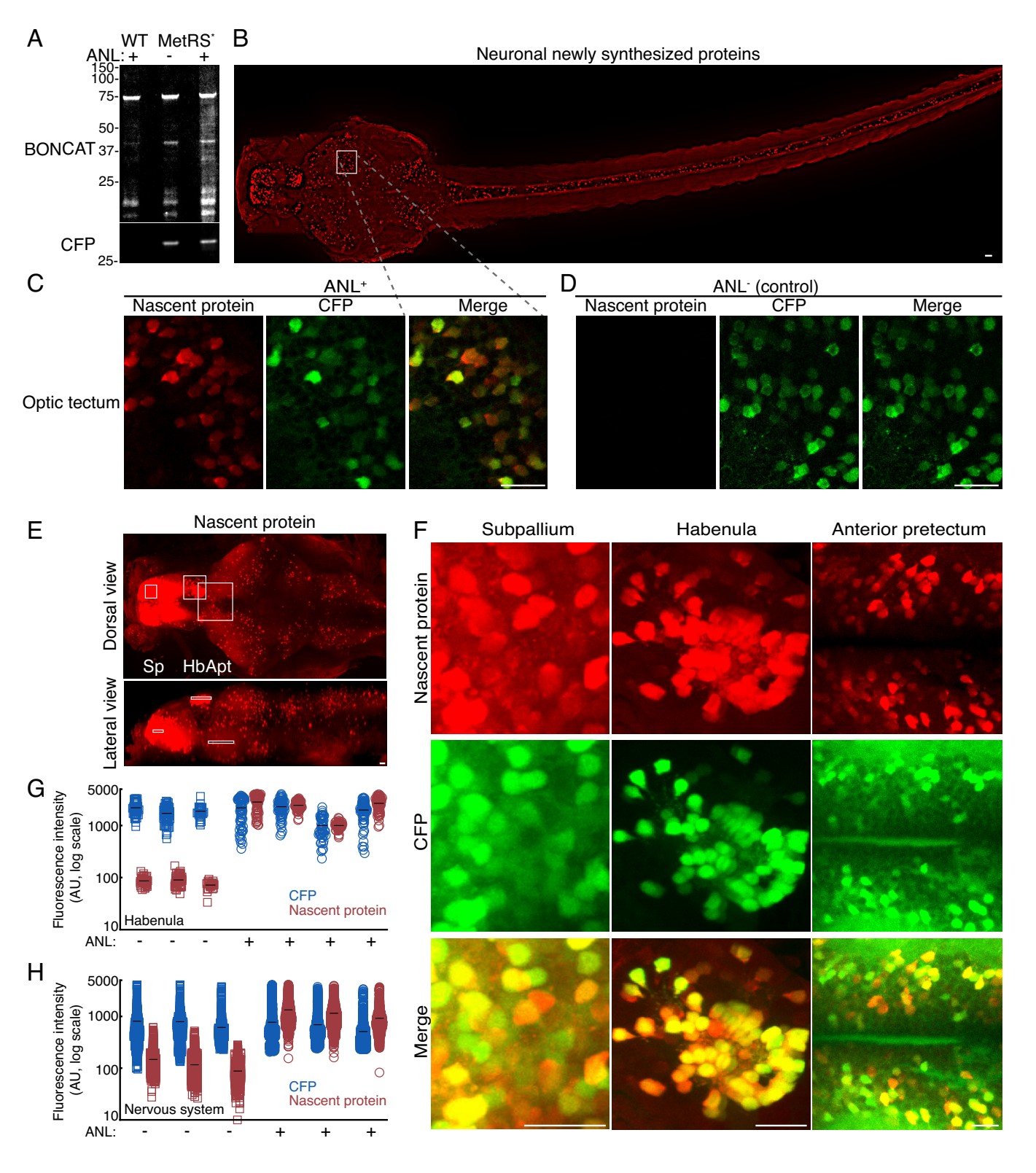

**Figure 2.** High resolution imaging newly synthesized proteins in different neuronal populations. 3 dpf larvae were incubated with 10 mM ANL for 24 hr in vivo, fixed (at 4 dpf) and clicked to a biotin tag for BONCAT or a fluorescent tag to visualize nascent proteins in situ. (**A**) Immunoblot detecting newly synthesized proteins in WT larvae treated with ANL (WT ANL+, control), MetRS$^{L270G}$ (MerRS*) larvae treated with ANL (ANL +) or not (ANL-, control). (**B**) Dorsal view collage projection of confocal images showing fluorescently labeled newly synthesized neuronal proteins (red). (**C-F**) High magnification

*Figure 2 continued on next page*

*Figure 2 continued*

view of different brain regions. 4–6 confocal planes are shown (~10 microns in depth). Note the overlap between the CFP channel (Ab staining, green) and the nascent protein channel (click labeling, red), indicating that the signal is specific to cells expressing the MetRS[L270G]. (C-D) Optic tectum (Ot). (C) Shown are 4 planes of the region indicated by the square in B. (D) The same region in a larva not incubated with ANL demonstrating the CFP but not nascent protein labeling. (E) Maximal projection of labelled newly synthesized proteins in an entire brain (dorsal view and lateral view) (see *Figure 2—figure supplement 2A–B* for lower brightness). White frames indicate the subpallium (Sp), habenula (Hb) and anterior pretectum (Apt), regions zoomed in (in F). (F) CFP Ab staining and nascent protein labeling in 4–6 confocal images indicated in the white frames in E. See *Figure 2—figure supplement 2* for more brain regions. (G-H) Quantification of the average nascent protein levels in the habenula (G) and the entire nervous system (H). Neurons were segmented in 3D using the CFP channel (see supplementary material and *Figure 2—figure supplements 6–7*) and the average voxel fluorescence intensity for the CFP and fluorescently labeled nascent protein was measured in each cell. Plotted are the average fluorescence intensities in single cells. 30 to 60 neurons were segmented for each habenula of 4 ANL-treated and 3 control larvae (G). (H) Quantification of the average CFP and nascent protein fluorescence intensity in neurons across the entire nervous system. More than 1000 neurons were segmented in 3D using the CFP channel (similar to G). Plotted are the mean fluorescence intensities in single cells from 3 larvae treated (ANL+, squares) or not treated (ANL-, circles) with ANL. See *Figure 2—figure supplement 5* for statistical differences between the groups. Squares: ANL⁻ (control), circles: ANL⁺, blue: CFP, red: Nascent protein, black line – mean fluorescence intensity within a single larva. One cell had a nascent protein intensity value below ten and is shown on the x-axis. Scale bars = 20 μm.

The online version of this article includes the following video, source data, and figure supplement(s) for figure 2:

**Source data 1.** Source data for *Figure 2G–H*.
**Figure supplement 1.** Quantification of Immunoblot following BONCAT for *Figure 2A*.
**Figure supplement 2.** Fluorescently labeled nascent proteins in neurons across the zebrafish larva brain.
**Figure supplement 3.** High resolution imaging of neuron-specific newly synthesized proteins in different brain regions – WT larvae exposed to 10 mM ANL for 24 hr (controls).
**Figure supplement 4.** Neuron-specific high resolution imaging of newly synthesized proteins in different brain regions.
**Figure supplement 5.** Quantification of neuronal nascent protein levels following 24 hr incubation with ANL.
**Figure supplement 6.** Three-dimensional segmentation of CFP positive cells.
**Figure supplement 7.** Three-dimensional segmentation of CFP positive cells in the habenula region.
**Figure supplement 8.** Detection of nascent protein in neuronal processes.
**Figure 2—video 1.** Shown are 3D projections of a representative region-of-interest (tegmentum or anterior hindbrain, see *Figure 2—figure supplement 6*) from the brain of an ELAVL3-MetRS[L270G] larva, demonstrating the segmentation of neurons using the CFP channel.
https://elifesciences.org/articles/50564#fig2video1

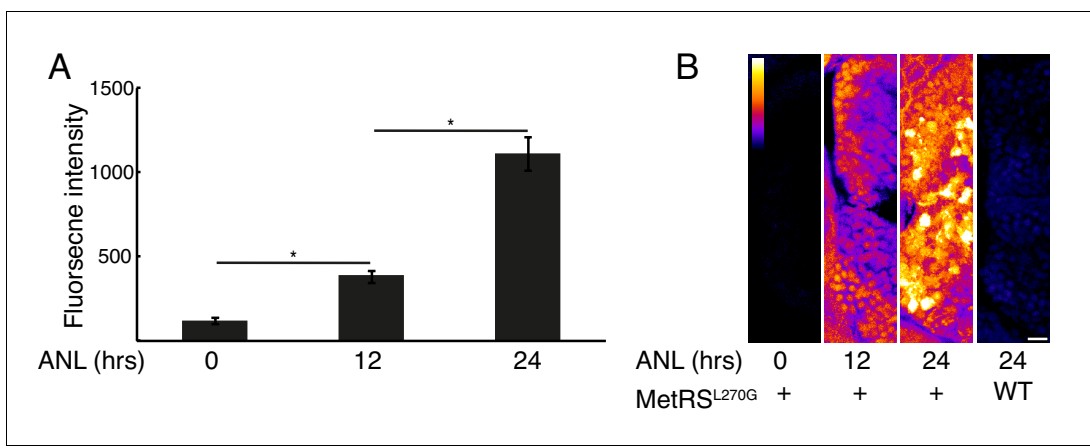

**Figure 3.** Newly synthesized proteins following different durations of metabolic labeling. Larvae were incubated with 10 mM ANL in their water bath for the indicated durations before fixation and click reaction. (A) Neurons in the habenula of MetRS[L270G] larvae were segmented in 3D using the CFP channel and the fnascent proteinintensity was measured in each neuron. Plotted are the average intensities of 3 larvae for each treatment (N = 3 for each treatment). Error bars indicate the SEM (N = 3). *p<0.05 (0.005 for 12 hr and 0.009 for 24 hr). (B) Representative images. Shown are maximal projections of 10 planes (~4 μm) of confocal images in the nascent proteins channel of the forebrain focusing on the olfactory bulb. Calibration bar – top left, scale bar = 10 μm.

The online version of this article includes the following source data for figure 3:

**Source data 1.** Source data for *Figure 3A*.

the larvae (at 4 dpf) and performed a whole-mount click reaction using a fluorescent alkyne tag. We performed immunostaining for CFP to visualize neurons expressing CFP (and therefore MetRS[L270G]) in the same larvae. We then imaged the entire nervous system visualizing both nascent protein and CFP (*Figure 2*, *Figure 2—figure supplements 2–5*). We detected nascent protein signal across the nervous system including in the subpallium, habenula, anterior pretectum, optic tectum, hindbrain, medulla and the spinal cord (*Figure 2*, *Figure 2—figure supplement 2*). In order to quantify the nascent protein signal, we segmented the cell somata volumes using the 3D CFP signal (*Figure 2—figure supplements 6–7* and *Figure 2—video 1*). We then measured the average voxel intensity in the fluorescent-alkyne channel to determine the level of nascent protein in each cell. We segmented hundreds to over two thousand cells in each larva. We calculated the average signal intensity in a specific brain region, the habenula (*Figure 2G*), and in the entire nervous system (*Figure 2H*). The same click reaction performed on larvae that were not incubated with ANL revealed only low levels of background fluorescence (*Figure 2C–D,G–H*). In some neurons, we could detect a nascent protein signal in neurites indicating the sensitivity of the method to visualize newly synthesized proteins in dendrites or axons (*Figure 2—figure supplement 2C–D*, *Figure 2—figure supplement 8*). These newly synthesized proteins could have either been synthesized in somata and moved to the processes, or could have been synthesized locally in the processes.

## Detecting nascent proteins with different periods of labeling

We found that 24 hr of ANL exposure was sufficient to allow the detection of nascent protein signal (*Figure 2*). In order to determine if an even shorter incubation period would result in labeling, we incubated 3 dpf zebrafish larvae for 12 hr with the same ANL concentration (10 mM). Quantification of the average nascent protein intensity in neurons revealed significant labeling compared to the control, but these levels were significantly lower than levels observed following 24 hr of ANL incubation (*Figure 3*). This relatively short labeling window could thus be used for comparing protein synthesis intensities between experimental and control fish in various behavioral paradigms.

## Detection of altered neuronal protein synthesis levels following seizures

We next addressed whether the nascent protein signal was sensitive to global alterations in neural activity. The GABAergic receptor antagonist Pentylenetetrazole (PTZ) induces epileptic-like neuronal discharges and seizure-like behaviors in rodents and zebrafish (*Baraban et al., 2005*; *Baraban et al., 2007*; *Naumann et al., 2010*). PTZ has been shown to induce expression of immediate early genes in larval zebrafish (*Baraban et al., 2005*). To determine the effect of elevated neural activity and behavioral seizures on protein synthesis levels in neurons in vivo, we incubated 3 dpf MetRS[L270G] larvae with ANL for 12 hr and induced seizures by adding PTZ for the last 2.5 hr (*Figure 4A*). We then measured neuronal protein synthesis levels within single cells in two different brain regions, the spinal cord and the habenula (*Figure 4B–E*). We segmented tens to hundreds of neurons in the spinal cord or right habenula and calculated the average protein synthesis signal in each cell (*Figure 4D*). Following PTZ exposure, we detected a significant (~60%) increase in protein synthesis levels in the habenula and a modest, though not significant, increase in the spinal cord (*Figure 4D–E*) when calculating the average intensity between several larvae.

## Discussion

We have developed a system that enables the labeling of nascent proteins in living zebrafish larvae in a cell-type-specific manner. We describe a UAS line that allows one to tag newly synthesized endogenous proteins in a cell type-of-interest simply by crossing it with any specific Gal4 driver line.

   Given the importance of protein synthesis for many biological processes, labeling nascent proteins for imaging within the intact organism will be useful for future studies. Non-canonical amino acids have been used to elucidate different biological processes including protein turnover (*Cohen et al., 2013*), protein dynamics (*Zhang et al., 2010*; *Schanzenbächer et al., 2016*) and local protein synthesis (*Dieterich et al., 2006*; *Tcherkezian et al., 2010*; *Yoon et al., 2012*). We have previously used AHA to label nascent proteins in zebrafish larvae (*Hinz et al., 2012*) without cell type specificity. To achieve cell-type-specific labeling in mice and other species, we have mutated the MetRS in the evolutionarily well-conserved methionine-binding pocket, resulting in cell type

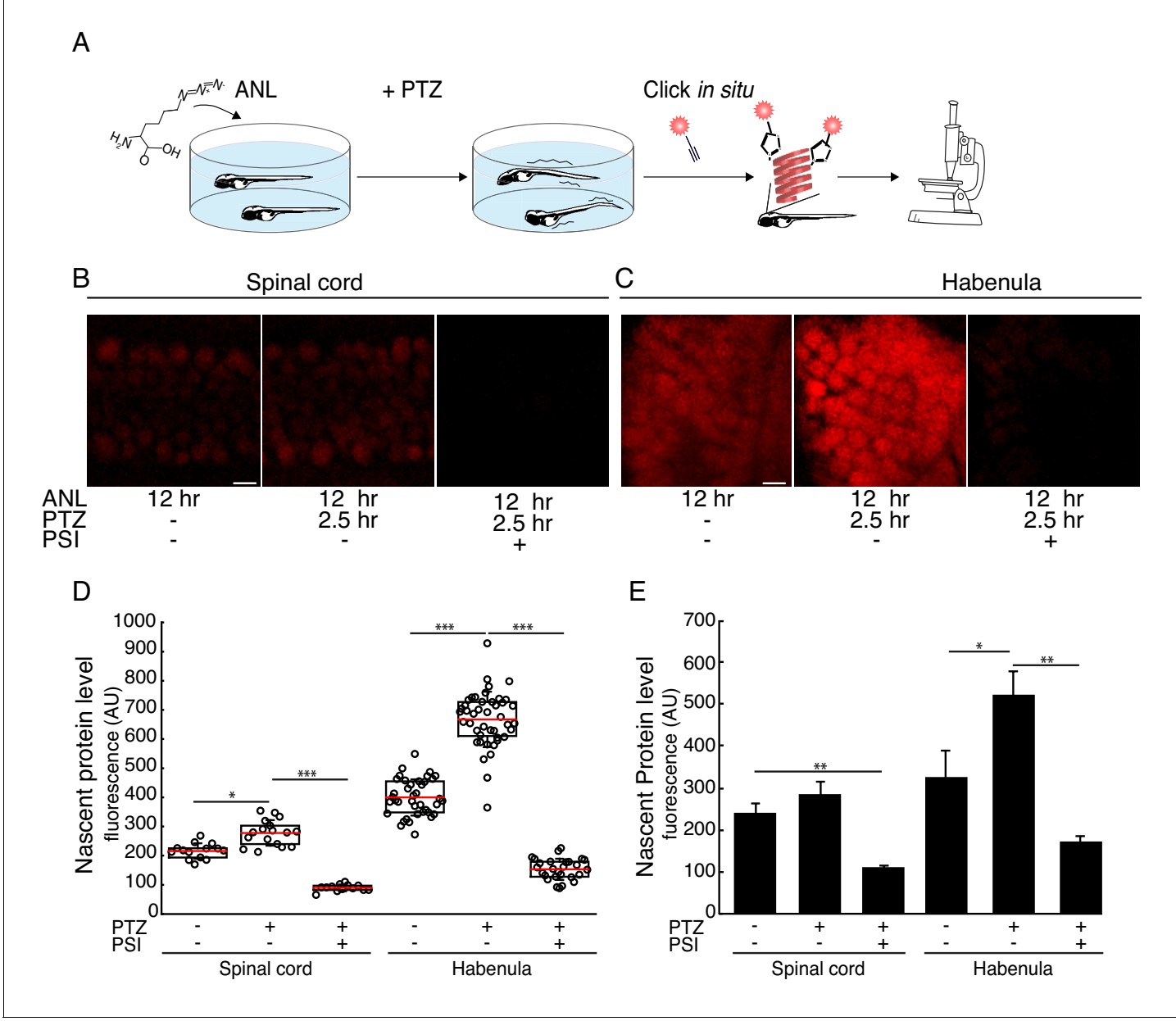

**Figure 4.** Seizure-induced neuron-specific protein synthesis. (**A**) Schematic of the experiment. Freely swimming larvae were incubated with ANL. After 10 hr incubation, PTZ was added for 2.5 hr, inducing seizures. Following fixation, whole-mount click with a fluorescent alkyne and confocal imaging were performed. (**B-C**) Representative images of the ANL signal in the spinal cord (**B**) or habenula (**C**). Shown are maximum projections of 4 confocal planes of the specific regions of larvae treated or non-treated with the protein synthesis inhibitor puromycin (PSI) and treated or non-treated with PTZ. (**D**) Quantification of the images shown in (**B-C**). Cells were segmented in 3D using the CFP channel, and the mean nascent protein labeling was measured in each cell using the fluorescent-alkyne channel. The dots represent the mean intensity in cells in the corresponding image (of B-C respectively). Red line – mean, whiskers - STDEV. (**E**) Bar plot showing the average nascent protein intensity in 3 to 5 larvae PTZ- and PTZ+, respectively. CFP positive cells in the spinal cord or the habenula were segmented in 3D using the CFP antibody staining. The levels of newly synthesized proteins were measured using the fluorescent alkyne (similar to D). More than 100 neurons in the spinal cord and 30 neurons in the habenula were analyzed. The average intensity was calculated for each larva. Plotted are the averaged intensities for each treatment (N = 3 to 5 larvae for each treatment). Error bars indicate SEM. *$p<0.05$, **$p<0.01$***$p<0.001$. Scale bars = 5 μm (**B, C**).

The online version of this article includes the following source data and figure supplement(s) for figure 4:

**Source data 1.** Source data for *Figure 4E*.
**Figure supplement 1.** Seizure-induced neuron-specific protein noise levels in WT.

specificity of nascent protein labeling in vivo (*Erdmann et al., 2015*; *Mahdavi et al., 2016*; *Alvarez-Castelao et al., 2017*).

We detected CFP expression levels above background in all cells with ANL labeling (*Figure 2*). Generally, we observed a positive correlation between the intensity levels of CFP (a proxy for the MetRS[L270G] mutant expression) and nascent protein, but there were a few cells detected with a very low expression of CFP and high nascent protein intensity or vice versa (*Figure 2*). The general positive correlation between CFP and nascent protein is expected because CFP is synthesized within the cells of interest and its level of protein synthesis will likely reflect the general translational activity of the cell. Cases in which the expression of CFP is not proportional to the nascent protein labeling may be explained by the following: the CFP labeling intensity is a result of both expression and degradation. CFP may thus bedegraded to a greater or lesser extent in some neurons. Zebrafish larvae exhibit high levels of neural activity during swimming and other natural behaviors (*Chen et al., 2018*). Many neuronal activity-dependent regulatory processes use both protein synthesis and proteasome-dependent protein degradation (*Sutton and Schuman, 2006*; *Hinz et al., 2013*; *Langebeck-Jensen et al., 2019*; *Djakovic et al., 2009*; *Bingol and Schuman, 2006*) to remodel the proteome. Recently, evidence for the coordination of protein synthesis and proteasome dependent degradation has been observed, including the degradation of nascent proteins by a neuron-specific 20S membrane proteasome complex (for example *Ramachandran et al., 2018*). It is possible that dynamic translation and degradation processes result in diverging levels of CFP and other proteins.

The fact that we detect neurons in which there is strong intensity of nascent proteins with only a faint CFP signal may suggest that even a low amount of the MetRS[L270G] mutant is sufficient for ANL incorporation into nascent proteins. We observed variation in the nascent protein intensities between larvae and between cells within the same larva and even within the same region and between neighboring cells (*Figure 2*). This is consistent with previous findings visualizing fluorescently labeled newly synthesized proteins using the non-canonical amino acid AHA (*Liu and Cline, 2016*).

ANL incorporation into nascent proteins was detectable in situ following 12 hr of exposure in the swim water, circumventing the need for ANL administration through food or drink (*Erdmann et al., 2015*; *Alvarez-Castelao et al., 2017*), and allowing better control over the concentration and duration of ANL exposure. This relatively short time window opens opportunities to measure protein synthesis levels in many biological paradigms. We have tried shorter incubation durations but the signal was sparse, the signal-to noise was too low and the variability between cells and between different larvae was high, indicating that shorter incubation time is not sufficient to measure nascent protein levels under these conditions. As far as we know, the 12 hr duration demonstrated here is the shortest available time frame so far for cell-type-specific metabolic labeling.

We demonstrate that this method can be used to address biological questions by labeling nascent proteins in neurons during an induced seizure-like behavior. PTZ-induced seizure-like behavior in zebrafish larvae results in higher expression levels of immediately early genes (*Hinz et al., 2012*; *Baraban et al., 2005*). We have previously shown a general increase in protein synthesis, using the non-canonical amino acid AHA which can incorporate into nascent proteins in all cell types (*Hinz et al., 2012*), following exposure of zebrafish larvae to PTZ. However, it was not possible in that study to determine whether the increased protein synthesis signal arose from neurons or other cell types. Here, we used PTZ to induce seizures and revealed an increase in neuronal protein synthesis. Furthermore, the specific labeling of newly synthesized proteins in neurons reduces background signals and noise from other cell types, thereby allowing us to compare different brain regions and neuron groups. For example, the habenula is a highly conserved structure that has been implicated in decision-making (*Hikosaka, 2010*). We found a significant increase in protein synthesis in the habenula following PTZ-induced seizures. Our findings are consistent with an increase in immediately early gene expression following PTZ-induced seizures in the habenula as has been reported in rats (*Del Bel et al., 1998*).

The method described here is robust: ANL, which is incorporated into nascent proteins, and the fluorescent alkyne used for imaging form a strong covalent bond allowing for stringent washes and hence low background signal. While the click chemistry used here precludes live imaging, the ANL incorporation takes place in vivo while the larvae are freely swimming and behaving. Therefore, the duration of the labeling can be chosen according to the studied cell-type, the protein synthesis levels (and degradation), and the biological question. The ability to label newly synthesized proteins in

vivo for a pre-chosen duration and to then 'freeze' the result before imaging has advantages. Because of the strong stability of the fluorescently labeled newly synthesized proteins following the click reaction one can image the entire larvae, as demonstrated here. This is especially useful for labeling neural networks during dynamic physiological processes such as cumulative calcium influx activity (*Fosque et al., 2015*) or protein synthesis following neural activity (demonstrated here). Beyond the contribution to neuroscience, this platform can be adopted by crossing the UAS-MetRS$^{L270G}$ zebrafish line described here to any existing Gal4 driver line of interest.

Zebrafish larvae have the advantage of being both translucent and smaller than mammals, therefore allowing one to label and image nascent proteins in an entire intact brain or other tissues without the need to slice the tissue. Future experiments using cell-type-specific newly synthesized proteins in zebrafish could include investigating the loci of learning and memory formation. Protein synthesis is essential for the formation of many types of long-term memory (*Davis and Squire, 1984*; *Agranoff et al., 1966*; *Flexner et al., 1962*; *Hinz et al., 2013*; *Roberts et al., 2013*). The method described here, combined with zebrafish larva learning paradigms (reviewed in *Roberts et al., 2013*), will enable future studies to reveal the neural networks in which protein synthesis occurs during learning and memory formation. Therefore, methods combining zebrafish larva with cell-type-specific protein synthesis labeling will be widely applicable.

# Materials and methods

## Key resources table

| Reagent type (species) or resource | Designation | Source or reference | Identifiers | Additional information |
|---|---|---|---|---|
| Antibody | Rabbit polyclonal anti GFP | Invitrogen | Cat# A11122 | IF(1:600) |
| Antibody | Chicken polyclonal anti GFP | Aves | Cat# GFP 1010 | WB(1:1000) |
| Antibody | Donkey polyclonal anti-chicken IR680 | Licor | Cat# 926–68075 | WB(1:10000) |
| Antibody | Goat anti-rabbit IR800 | Licor | Cat# 925–32211 | WB(1:10000) |
| Zebrafish line | HuC-Gal4 | Stevenson, T. J. et al | | Schuman lab, See Materials and methods section |
| Zebrafish line | UAS-MetRS$^{L270}$ | Tefor-Amagen/ This paper | | Schuman lab, See Materials and methods section |
| Zebrafish line | ELAVL3-MetRS$^{L270G}$ | Tefor-Amagen/ This paper | | Schuman lab, See Materials and methods section |

## Zebrafish husbandry

Adult fish strains AB, UAS-MetRS$^{L270}$, ELAVL3-MetRS$^{L270G}$ and HuC-Gal4 were kept at 28°**C** on a 14 hr light/10 hr dark cycle, in a Techinplast Zebtec system. Embryos were obtained from natural spawning using a Techinplast breeding system, and were maintained in E3 embryo medium (5 mM NaCl, 0.17 mM KCl, 0.33 mM CaCl2, 0.33 mM MgSO4) at 28°**C** on a 14 hr light/10 hr dark cycle.

## Constructs and transgenic zebrafish

The following plasmid was injected to wt (AB strain) eggs at one cell stage: pBT2_4xnr UAS-CFP (Tol2-4xnr UAS:cerulean-2A-MetRS-6x His –Tol2). MetRS (Methionyl-tRNA synthetase (AAH57463.1) *Danio rerio* with mutation to L270G. The 4x non-repetitive (nr) UAS sequence was designed by the Halpern lab (*Akitake et al., 2011*). The following primers were used for genotyping: forward-

gcaagggcgaggagctg, reverse: gctcaggtagtggttgtcg. The PCR product size was 602 bp. Huc:Gal4 was generated by the Piotrowski lab (*Stevenson et al., 2012*).

## ANL administration

Azidonorleucine (ANL) was synthesized as previously described *Mahdavi et al. (2016)*. ANL was kept as powder and was freshly dissolved in E3 solution prior to experiments at the indicated concentrations (0, 5, 10 and 20 mM). For the dosage determination experiment, Konstanz wt or ELAVL3-MetRS$^{L270G}$ freely swimming zebrafish larvae were supplemented with ANL or mock E3 exchange for 24 hr. During the last 20 min, larvae were moved to a chamber with swimming lanes that allowed single larvae to freely swim. The positions of the larvae were recorded with a camera at 1 Hz, and tracked automatically with a Matlab script to measure swimming distance and speed. In all the fluorescent labeling experiments, the ANL concentration was 10 mM. 0.1 mM 1-phenyl 2-thiourea (PTU) was added to the E3 water at 1 dpf for all larvae in *Figures 2–4*.

## Click chemistry for fluorescent tagging

3–4 dpf old larvae were anaesthetized on ice for 45 min. Larvae were transferred to 1.5 ml tubes and washed once with clean E3. E3 was removed and replaced with 1 ml fixation solution (4% PFA, 4% Sucrose in PBS (137 mM NaCl, 2.7 mM KCl, 4.3 mM Na2HPO4, 1.4 mM KH2PO4)), and incubated overnight at 4°C with gentle shaking. Next, larvae were dehydrated in Methanol at −20°C overnight. Samples were gradually rehydrated through successive 5 min washes with 75% methanol in PBST (PBS+0.1% Tween-20), 50% methanol in PBST, 25% methanol in PBST and PBST. Samples were incubated 3 times for 5 min in PBDTT (PBST + 1% DMSO and 0.5% Triton X-100) and digested with 1 mg/ml collagenase (Sigma-Aldrich) in PBST for 45 min for permeabilization. Following one wash with PBST, larvae were post-fixed for 20 min in 4% PFA and 4% Sucrose in PBS. Samples were washed 3 times for 5 min in PBDTT and incubated for 3 hr at 4°C in blocking buffer (5% BSA, 10% goat serum in PBDTT). Samples were then washed 3 times for at least 10 min in PBST (pH 7.8). ANL labeled proteins were tagged using a click reaction. To 1 ml PBS (pH 7.8) TBTA (Sigma-Aldrich) 1:500 (stock 200 mM in DMSO, final 0.2 mM) was added followed by a 10 s vortex, TCEP (Thermo Scientific) 1:400 (stock 40 mM in H$_2$O, final 0.5 mM), 10 s vortex, 1:500 AlexaFluor-647-alkyne (Invitrogen, stock 2 mM in DMSO, final 2 μM), 10 s vortex, and 1:500 CuSO$_4$ (stock 200 mM in H$_2$O, final 0.2 mM) followed by 30 s vortex. The above click reaction buffer was immediately added to tubes containing the larvae and kept overnight in the dark, at 4°C with gentle agitation. Samples were washed 4 times for 30 min in PBDTT with 0.5 mM EDTA and twice for one hour in PBDTT and kept in PBS (pH 7.4) until mounting. For immunofluorescence, samples were incubated in blocking buffer (10% serum in PBS) for 1 hr and incubated with 1:600 rabbit anti GFP antibody (Invitrogen A11122; to detect CFP) in blocking buffer overnight with gentle agitation at 4°C. Samples were washed in PBST twice and PBS 3 times (~10 min each wash), incubated with blocking buffer for ten minutes and incubated with an Alexa-488 fluorescent secondary antibody (goat anti rabbit, ThermoFisher A1008) overnight with gentle agitation at 4°C. Samples were washed twice with PBST and PBS (pH 7.4). After click or immunofluorescence, the samples were gradually moved to glycerol through successive 5 min washes (25% glycerol in PBS, 50% glycerol in PBS, 75% glycerol in PBS) and finally 100% glycerol and kept at 4°C in dark.

## BONCAT

Larvae were incubated with 10 mM ANL for 24 hr. After incubation, the media was replaced with fresh E3. Larvae were sacrificed on ice-cold water (30 min). Heads were dissected using a scalpel and immediately snap frozen in tubes (1.5 ml, Eppendorf) that were pre-cooled using dry ice and stored at −80C until lysis. Tissue was homogenized and lysed using a pestle in lysis buffer (1% in Triton X100, 0.4% (w/v) SDS in PBS pH 7.8, 1:1000 EDTA free protease inhibitors (Calbiochem) and benzonase (Sigma, 1:1,000), and denatured at 75°C, 13,000 rpm for 10 min. Lysates were then cleared by centrifugation. BONCAT was performed as previously described (*Dieterich et al., 2007*). In brief, 60 μg proteins were dissolved in 120 μL PBS pH 7.8 supplemented with 0.01% SDS, 0.1% Triton, 300 μM Triazol (Sigma, 678937), 50 μM biotin-alkyne tag (Thermo, B10185) and 83 μg/mL CuBr (prepared by dilution of fresh 10 mg/mL solution in DMSO) at 4°C overnight in the dark. Biotinylated proteins were then separated by gel electrophoresis and immunoblotted with 1:1000

chicken anti-GFP (Aves), 1:1000 rabbit anti-biotin (Cell signaling) and Donkey anti-chicken IR680, goat anti-rabbit IR800 (IB, 1:10,000, Licor) antibodies.

## Microscopy

Larvae were directly, or after dissection to remove the eyes and yolk sack (*Figures 2–4*), mounted on Mattek dishes with the dorsal side facing the glass bottom of the dish prior to imaging. Larvae were imaged using a Zeiss LSM880 confocal microscope and a 25X glycerol objective (NA 0.8, PSF: X 0.389, Y 0.336, Z 1.62). Spacing between z planes = 1.89 micron. 488 and 633 lasers were used for Alexa488 and Alexa647, respectively (filters: 504–563 and 652–702, respectively). To image the entire brain, tiling was done with 10% overlap using the ZEN program with a scaling of 0.332 micron per pixel and a 12-bit mode; tiles were stitched together using the ZEN program after imaging. In *Figure 2*, in order to allow for the clear visualization of 3-dimensional data of an entire larva in 2 dimensions, a collage was made to exclude non-relevant planes (e.g: in each region of the tail, only 3–5 planes out of ~150 contain neurons, therefore a maximal projection of the entire stack would impede a clear visualization of the neurons).

## 3D fluorescence intensity analysis

Stitched images were imported to Imaris 9.2. Cell segmentation was conducted using CFP channel intensities. Surface segmentation of cells in the entire image was based on the Marching cube algorithm (*Lorensen and Cline, 1987*) using the Imaris program. In some regions where we noticed that the segmentation did not perform well, a regional segmentation was performed again using the same algorithm with a manual threshold in smaller specific ROIs. False positive objects that were not real cells, such as salt crystals, were deleted based on morphology. Splitting of touching objects was allowed using the watershed algorithm. In some cases where splitting of touching objects failed due to a high density of neurons, a few cells were inevitably considered as one. The threshold of the minimal volume was set to 10 $\mu m^3$, and of the minimal 'sphericity' was set to 0.75. An example of segmentation in a low density ROI can be found in *Figure 2—figure supplement 6*, and in *Figure 2— video 1*. Segmentation of a densely labeled ROI in the habenula is demonstrated in *Figure 2—figure supplement 7*. After segmentation in the CFP channel, the intensity values were measured in both the CFP and the ANL channels (for *Figure 2*) or for the ANL channel (*Figures 3–4*). For each cell, the average voxel intensity was calculated and documented. Next, based on the average cell intensities, the average intensity of cell in a region or an entire larva was calculated. Finally the mean intensity between 3 to 4 larvae was calculated for each region or entire larvae and plotted (n = 3 or 4). The error bars in all the graphs represent Standard Error of the Mean (SEM), unless otherwise noted. In *Figure 4D*, the bars represent SEM between cells. In all the other graphs, error bars represent SEM between the average intensities of several larvae. One-tailed, two-sample unequal variance (heteroscedastic) TTESTs were used for calculating all the p values (p).

## Acknowledgements

We thank Jan Gluesing and Anja Staab for technical assistance, Anett-Yvon Loos for fish husbandry and managment, Doug Campbell for plasmid design, Alejandro Pinzon-Olejua for help with line screening, Susanne tom Dieck for ANL synthesis and Florian Vollrath for image processing assistance. Work in the laboratory of EMS is supported by the Max Planck Society and The European Research Council under the European Union's Horizon 2020 research and innovation program (grant agreement No 743216). We also acknowledge the support of DFG CRC 902 and 1080. ODS was supported by an EMBO Long-Term Fellowship and by the European Union's Horizon 2020 research and innovation programme under the Marie Sklowdoska-Curie grant agreement 628003.

## Additional information

### Funding

| Funder | Grant reference number | Author |
| --- | --- | --- |
| European Molecular Biology Organization | ALTF-643-2014 | Or David Shahar |

| Seventh Framework Pro-gramme | Marie Curie 628003 | Or David Shahar |
| Max-Planck-Gesellschaft | Research grant | Erin Margaret Schuman |
| Horizon 2020 Framework Pro-gramme | 743216 | Erin Margaret Schuman |
| Deutsche Forschungsge-meinschaft | CRC 902 | Erin Margaret Schuman |
| Deutsche Forschungsge-meinschaft | CRC 1080 | Erin Margaret Schuman |

The funders had no role in study design, data collection and interpretation, or the decision to submit the work for publication.

### Author contributions
Or David Shahar, Conceptualization, Formal analysis, Investigation, Visualization, Methodology; Erin Margaret Schuman, Conceptualization, Supervision, Funding acquisition, Project administration

### Author ORCIDs
Or David Shahar https://orcid.org/0000-0002-5039-8307
Erin Margaret Schuman https://orcid.org/0000-0002-7053-1005

### Decision letter and Author response
Decision letter https://doi.org/10.7554/eLife.50564.sa1
Author response https://doi.org/10.7554/eLife.50564.sa2

## Additional files

### Supplementary files
• Transparent reporting form

### Data availability
All data generated or analyzed during this study are included in the manuscript and supporting files. Image data is available on the Max Planck Digital Library (https://doi.org/10.17617/1.8L).

The following dataset was generated:

| Author(s) | Year | Dataset title | Dataset URL | Database and Identifier |
|---|---|---|---|---|
| Shahar OD, Schuman EM | 2020 | Shahar2020 | https://doi.org/10.17617/1.8L | Max Planck Digital Library, 10.17617/1.8L |

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
