## [Decision Letter]

**Acceptance summary:**

This article describes a tool for marking newly synthesized proteins in a cell-type specific manner in larval zebrafish. Such a tool would likely be of use to mark nascent proteins in a cell-type specific or tissue-specific manner in order to understand how novel protein synthesis is involved in development or disease for example.

**Decision letter after peer review:**

[Editors’ note: the authors submitted for reconsideration following the decision after peer review. What follows is the decision letter after the first round of review.]

Thank you for submitting your work entitled "Large scale cell-type-specific imaging of protein synthesis in a vertebrate brain" for consideration by *eLife*. Your article has been reviewed by three peer reviewers, and the evaluation has been overseen by a Reviewing Editor and a Senior Editor. The reviewers have opted to remain anonymous.

Our decision has been reached after consultation between the reviewers. Based on these discussions and the individual reviews below, we regret to inform you that your work will not be considered further for publication in *eLife*.

While all three reviewers agreed that this tool was useful for the zebrafish field and could be easily adopted by investigators, the fact that this method has already been established in other model organisms and is an extension of previous work diminished the overall novelty and significance of this tool. In addition, reviewers also had suggestions for improving demonstration of specificity of this tool (see full comments below).

Reviewer #1:

Shahar and Schuman apply a bio-orthogonal approach to study the dynamics and extent of protein synthesis in a cell type specific manner in zebrafish. By driving MetRSL270G expression under the ELAVL3 neuron-specific promoter, and adding the unnatural amino acid, azidonorleucine (ANL) to the swim water, the authors observe ANL incorporation indicative of new protein synthesis in neurons in the heterogenous environment of the brain and spinal cord. Using this approach. They then show that protein expression is increased upon seizure induction using the GAG receptor antagonist, PTZ. The authors suggest that the approach can, in principle, be extended to other tissues and cell types and questions, with the UAS line generated serving as a tool for the community.

Previously, the Schuman lab showed that it is possible to examine protein synthesis using a similar approach in whole larvae (Hinz et al., 2011). They now extend their previous findings to a tissue-specific manner (i.e larval brain and spinal cord neurons) and which (as they report in the Introduction) has been shown previously in other organisms. The Gal4 UAS system for cell type specific expression has been used in zebrafish for a long time. The ability to image sites of protein synthesis in the brain of intact animals post behaviour is likely to be of use to identify regions of the brain important for behaviour (- which is currently visualised in vivo by large through calcium imaging). Overall, the manuscript is straightforward and easy to understand. However, it does not represent a significant advance.

– The authors should examine ANL incorporation upon inhibition of protein synthesis using cycloheximide or other protein synthesis inhibitor. This will determine if the labelling is specific.

– The 12 hour incorporation window for the UNAA appears rather long in the context of a fast developing zebrafish larva. What is the half -life of ANL? And what is the shortest ANL exposure window within which incorporation can be detected. Though the 2 hour PTZ treatment shows clear differences over control larvae, a shorter ANL exposure might decrease noise from protein synthesis.

– It would be helpful to discuss the advantages/disadvantages of this approach as compared to other approaches to estimate newly synthesized proteins.

– Figure 4 graph D is specific for the animal shown in images B and C, but does not appear to be representative for all the animals analysed, as shown in graph E. It would be useful to do this experiment without ANL and with ANL in wild type controls to examine background autofluorescence or non-specific click reactions post PTZ incubation.

Reviewer #2:

Or and Schuman describe a method for cell type specific evaluation of new protein synthesis in intact Zebrafish using transgene expression of a mutant MetRS, that is modified to only charge the non-canonical amino acid, azidonorleucine (ANL). This is a valuable method that will likely be widely used in the Zebrafish community. The method and analysis are described in sufficient detail to allow most investigators to apply it successfully.

Addressing the following points would strengthen the paper.

1) Previous studies in *Drosophila* suggested that expression of mutant MetRS during development was detrimental. Does use of HuC-gal4 address this problem? In addition to swimming speed, the authors should test behaviors indicative of brain function, such as sensory information processing, to address the concern of developmental toxicity.

2) The data (for instance Figure 2G) indicate that NSP labeling is not consistently proportional to CFP levels, as a proxy for mutant MetRS expression. Although there are many potential reasons for this, the authors should comment on this potentially interesting observation, and in particular mention variation in rates of protein synthesis and rapid turnover of NSPs within the 24h period examined.

3) It is not correct to refer to 'the intensity of protein synthesis' since NSP lifetimes can be <24h.

4) Similarly 'averaged translation intensity' is not an accurate term.

5) Can the authors comment on the efficiency of ANL labeling NSPs compared to other available methods?

6) Is application of this method limited to Zebrafish larvae or can is be used in older animals? Does ANL penetrance to the brain change over larval development?

7) The text and figure legends differ in statements about use 3 or 4 dpf larvae.

8) The subsection “Induced seizures result in higher protein synthesis levels in neurons” refers to 2.5 h in PTZ. Figure 4A legend and schematic say 2 h.

9) Subsection “Detection of endogenous nascent proteins in neurons across the brain”. The FUNCAT label in neuronal processes could be NSPs that were synthesized in somata and transported into processes. It is not accurate to claim demonstration of sensitivity to detect dendritic protein synthesis in vivo.

10) The demonstration of the cell body segmentation in sparsely labeled brain is very nice, but segmentation using Imaris is challenging in densely labeled regions like the habenula. Can you include an example of the segmented images from habenula?

11) Discussion, third paragraph. The observation that PTZ induces NSPs in habenula in Zebrafish is consistent with the previous study by Del Bel et al. in rats, not the other way around.

12) Subsection “Zebrafish husbandry”. For non-Zebrafish aficionados, it would be helpful to provide more information on the breeding strategy. For instance, how were the HuC-Gal4 and nacre animals used?

13) Subsection “Constructs and transgenic zebrafish”. What are 'AB' eggs?

14) Subsection “BONCAT”. 'pistil' should be 'pestle'?

15) Figure 1G. The black images suggest extreme contrast settings.

16) Figure 2. Figure panel labels don't all match text in figure legend.

17) Figure 4D. Explain the box and whiskers and red line. Are the data plotted from a single animal, only from images shown in B and C?

18) The Liu and Cline paper showing cell type analysis of AHA labeling should be cited.

19) The Schanzenbacher paper is not listed numerically in the references.

Reviewer #3:

In this paper, Shahar and Schuman report a cell-type specific method for labeling newly synthesized proteins in zebrafish larvae. The method relies on expressing a mutated Methionyl tRNA synthetase in neurons using the well-established elav promoter. The mutated enzyme loads the unnatural amino acid azidonorleucine (ANL) instead of methionine into polypeptides. ANL can then be visualized using click-chemistry with a fluorescent alkyne. Using this protocol, authors show brain wide labeling in neurons and increase in intensity following chemically-induced seizures. As mentioned by the authors in the Introduction, cell-type specific labeling of new proteins has been shown before in *C. elegans, Drosophila* and mice and its establishment in zebrafish larvae is not a major advancement. All of the images in the manuscript are of fixed larvae. Given the obvious advantages of zebrafish for in vivo imaging, the paper would have been exciting if the authors were able to track novel protein synthesis in live animals. But the permeabilization steps required for click-labeling preclude such a possibility. There are other possibilities too such as cell-type specific proteomics characterization of the ANL-labeled proteins but this manuscript does not go that far.

In addition, the authors could address the following to make the manuscript stronger:

1) The authors will do well to establish better the specificity of this tool by showing more control data. Figure 2A shows that there is some labeling even in wild type. Authors need to show the corresponding fluorescence image (WT+ANL+) in Figure 1G. WT+ANL+ control data must also be added to Figure 3. Additionally, even without ANL addition, there is some labeling (Figure 2A, middle lane), probably corresponding to non-specific binding of the fluorescent tag. Thus, not all the signal observed corresponds to novel protein synthesis. A quantification of this non-specific background is required.

2) Related to the point above, were the ANL- samples in Figures 1 and 2 processed for click-dye labeling? Why are the images completely dark?

3) In Figure 2G, authors show colocalization of CFP and click-label in confocal stacks made from 4-6 planes spanning several cell layer thickness. Single optical slices need to be shown.

4) Authors report average of averages in Figures 2H and 2I. As such, the variability in signal amplitude seen in different neurons will likely be much higher than the spread shown. It is not clear how much variability is seen within one region, say habenula, in a single larva. It will also be better to show the data scatter for the bar plots shown in 2H and 2I. This is now standard practice.

5) Why were the spinal cord and habenula alone chosen for the PTZ experiment?

6) Authors call data in Figure 4D and E "Protein Synthesis Level". This is too strong – the signal likely reflects NET incorporation of ANL deriving from both synthesis and degradation of proteins.

7) Supplementary Figure 3 does not seem to be referred anywhere in the text.

8) There are several places where the figure panels and their description in legends or the Results section are mismatched.

---

## [Author Response]

[Editors’ note: The authors appealed the original decision. What follows is the authors’ response to the first round of review.]

Reviewer #1:
*[…] Previously, the Schuman lab showed that it is possible to examine protein synthesis using a similar approach in whole larvae (Hinz* et al.*, 2011). They now extend their previous findings to a tissue-specific manner (i.e. larval brain and spinal cord neurons) and which (as they report in the Introduction) has been shown previously in other organisms. The Gal4 UAS system for cell type specific expression has been used in zebrafish for a long time. The ability to image sites of protein synthesis in the brain of intact animals post behaviour is likely to be of use to identify regions of the brain important for behaviour (- which is currently visualised* in vivo by large through calcium imaging). Overall, the manuscript is straightforward and easy to understand. However, it does not represent a significant advance.– The authors should examine ANL incorporation upon inhibition of protein synthesis using cycloheximide or other protein synthesis inhibitor. This will determine if the labelling is specific.

We added data showing the significant inhibition of the nascent protein signal when the protein synthesis inhibitor Puromycin was added. We note that inhibition was also observed in the PTZ (seizure-induction) experiment where very high levels of protein synthesis were detected in the PTZ-treated condition. See revised Figure 4.

– The 12 hour incorporation window for the UNAA appears rather long in the context of a fast developing zebrafish larva. What is the half -life of ANL? And what is the shortest ANL exposure window within which incorporation can be detected. Though the 2 hour PTZ treatment shows clear differences over control larvae, a shorter ANL exposure might decrease noise from protein synthesis.

There is no data regarding the half-life of ANL. We have tried shorter incubation times but the signal was sparse, the signal-to-noise was weak and the variability between cells within a larva and between different larvae was high, indicating that under current experimental conditions a short incubation time is not sufficient to measure nascent protein levels. The 12-hour duration demonstrated in this manuscript is the shortest labelling period reported so far for in vivo cell-type-specific metabolic labeling.

– It would be helpful to discuss the advantages/disadvantages of this approach as compared to other approaches to estimate newly synthesized proteins.

We have now added paragraphs in the Introduction (last paragraph) and in the Discussion (last two paragraphs) describing the advantages and disadvantages of the method as compared to other approaches.

– Figure 4 graph D is specific for the animal shown in images B and C, but does not appear to be representative for all the animals analysed, as shown in graph E. It would be useful to do this experiment without ANL and with ANL in wild type controls to examine background autofluorescence or non-specific click reactions post PTZ incubation.

We have done these control experiments and added them to the revised manuscript: Figure 4D, E now contains controls with protein synthesis inhibitor (PSI). We believe that the ANL+/PSI+ control clarifies the background signal in the mutant line, including the remaining non-specific signal (after the washes) and the signal of non-specific click reactions. Additionally, we added a supplementary figure for Figure 4 with WT larvae incubated with ANL and PTZ and subjected to the FUNCAT protocol together with the samples shown in Figure 4. We did not detect a significant signal above background. We note that the WT + ANL control experiments are shown in Figure 4—figure supplement 1.

Reviewer #2:[…] Addressing the following points would strengthen the paper.1) Previous studies in *Drosophila* suggested that expression of mutant MetRS during development was detrimental. Does use of HuC-gal4 address this problem? In addition to swimming speed, the authors should test behaviors indicative of brain function, such as sensory information processing, to address the concern of developmental toxicity.

We did not see any detrimental developmental effects in larvae or fish expressing the MetRS mutant. We have already more than 3 generations of both the UAS-MetRS^L270G^ and then ELAVL3- MetRS^L270G^ lines and we even have them raised in other laboratories for future collaborations. In addition, to the swim speed data in the original submission, we have now added additional behavioral data regarding the intact light preference of larvae. Furthermore, we detect active swimming of freely swimming larvae according to their natural preference in a chamber upon frequent light/dark changes in defined regions within the chamber. Please see the new Figure 1—figure supplement 1. We use this assay routinely and we don’t see any difference between the transgenes to wt larvae (for example in Hinz FI, et al., 2013).

2) The data (for instance Figure 2G) indicate that NSP labeling is not consistently proportional to CFP levels, as a proxy for mutant MetRS expression. Although there are many potential reasons for this, the authors should comment on this potentially interesting observation, and in particular mention variation in rates of protein synthesis and rapid turnover of NSPs within the 24h period examined.

We agree with the comment and now add the explanations for both the general correlation between the CFP\MetRS* levels to the nascent protein intensities as well as for the lack of consistency in some of the cells. Please see the revised Discussion and specifically the third paragraph.

3) It is not correct to refer to 'the intensity of protein synthesis' since NSP lifetimes can be <24h.

Corrected.

4) Similarly 'averaged translation intensity' is not an accurate term.

Corrected.

5) Can the authors comment on the efficiency of ANL labeling NSPs compared to other available methods?

We did not directly measure efficiency levels. Nevertheless, we thank the reviewer for this comment and we add a paragraph in the Discussion comparing the method to fluorescently labeled reporters. We also highlight that this method is the shortest labeling for cell-type-specific non canonical amino acid tagging in vivo.

6) Is application of this method limited to Zebrafish larvae or can is be used in older animals? Does ANL penetrance to the brain change over larval development?

We used the ELAVL3 promoter, which is an early pan-neuronal marker. We did so in order to be able to conduct imaging when the larvae are still translucent. We therefore did not test the method in older fish. This would require a different promoter and therefore a new fish line. As mentioned, the strength of the system is that it can be used for any promoter of interest and indeed it would be interesting to examine in future studies the incorporation of ANL at later ages and whether dissolving the ANL in the water is sufficient.

7) The text and figure legends differ in statements about use 3 or 4 dpf larvae.8) The subsection “Induced seizures result in higher protein synthesis levels in neurons” refers to 2.5 h in PTZ. Figure 4A legend and schematic say 2 h.

*9) Subsection “Detection of endogenous nascent proteins in neurons across the brain”. The FUNCAT label in neuronal processes could be NSPs that were synthesized in somata and transported into processes. It is not accurate to claim demonstration of sensitivity to detect dendritic protein synthesis* in vivo.

We clarified all the text issues and typos mentioned by the reviewer.

10) The demonstration of the cell body segmentation in sparsely labeled brain is very nice, but segmentation using Imaris is challenging in densely labeled regions like the habenula. Can you include an example of the segmented images from habenula?

We agree with the reviewer, that the segmentation is challenging and not always perfect but it was good for our goal. As requested, we added in the revised manuscript an example of the segmentation in the densely labeled habenula in Figure 2—figure supplement 7.

11) Discussion, third paragraph. The observation that PTZ induces NSPs in habenula in Zebrafish is consistent with the previous study by Del Bel et al. in rats, not the other way around.12) Subsection “Zebrafish husbandry”. For non-Zebrafish aficionados, it would be helpful to provide more information on the breeding strategy. For instance, how were the HuC-Gal4 and nacre animals used?13) Subsection “Constructs and transgenic zebrafish”. What are 'AB' eggs?14) Subsection “BONCAT”. 'pistil' should be 'pestle'?

All the text mistakes (in points 11 to 14) were corrected.

15) Figure 1G. The black images suggest extreme contrast settings.

All the images in Figure 1G were acquired with the same settings. These are maximal intensity projections of stacks in order to show a large portion of the brain. Therefore, we used settings that allows for the visualization of both controls and the positive sample. We understand the concern of the reviewer and we add in the revised manuscript a video with the raw data of this image including the controls, plane by plane along the z axis (please see Figure 1—videos 1 and 2).

16) Figure 2. Figure panel labels don't all match text in figure legend.

Corrected

17) Figure 4D. Explain the box and whiskers and red line. Are the data plotted from a single animal, only from images shown in B and C?

In Figure 4D, the data plotted is the analysis of the images shown in B, C. Each dot is one cell. The red line is the average of the cells seen in the images. The whiskers are the STDEV. In Figure 4E, each bar represents the average of several larvae. In each larva, the average intensity was calculated and then an average of the average is plotted. Bars are standard error of the mean (n=# larvae). This is explained in the revised text and figure legend.

18) The Liu and Cline paper showing cell type analysis of AHA labeling should be cited.19) The Schanzenbacher paper is not listed numerically in the references.

Technical formatting issues were corrected.

Reviewer #3:

*In this paper, Shahar and Schuman report a cell-type specific method for labeling newly synthesized proteins in zebrafish larvae. The method relies on expressing a mutated Methionyl tRNA synthetase in neurons using the well-established elav promoter. The mutated enzyme loads the unnatural amino acid azidonorleucine (ANL) instead of methionine into polypeptides. ANL can then be visualized using click-chemistry with a fluorescent alkyne. Using this protocol, authors show brain wide labeling in neurons and increase in intensity following chemically-induced seizures. As mentioned by the authors in the Introduction, cell-type specific labeling of new proteins has been shown before in C. elegans, Drosophila and mice and its establishment in zebrafish larvae is not a major advancement. All of the images in the manuscript are of fixed larvae. Given the obvious advantages of zebrafish for* in vivo *imaging, the paper would have been exciting if the authors were able to track novel protein synthesis in live animals. But the permeabilization steps required for click-labeling preclude such a possibility. There are other possibilities too such as cell-type specific proteomics characterization of the ANL-labeled proteins but this manuscript does not go that far.*

Whereas the click chemistry precludes live imaging, the ANL incorporation is done in vivo while the larvae are freely swimming and behaving. The experimentalist can choose the duration of the labeling according to the studied cell-type, the protein synthesis levels (and degradation) and the biological question. The ability to label newly synthesized proteins in vivo for a prechosen duration and then “freeze” the result before imaging has advantages. Because of the strong stability of the fluorescently labeled newly synthesized proteins following the click reaction (which forms covalent bonds), one can image the entire larvae as we demonstrated. In the revised manuscript, we added paragraphs in the Discussion regarding the advantages and state of the art.

In addition, the authors could address the following to make the manuscript stronger:1) The authors will do well to establish better the specificity of this tool by showing more control data. Figure 2A shows that there is some labeling even in wild type. Authors need to show the corresponding fluorescence image (WT+ANL+) in Figure 1G. WT+ANL+ control data must also be added to Figure 3. Additionally, even without ANL addition, there is some labeling (Figure 2A, middle lane), probably corresponding to non-specific binding of the fluorescent tag. Thus, not all the signal observed corresponds to novel protein synthesis. A quantification of this non-specific background is required.

We performed the WT ANL+ control for all the experiments and now add the add images of WT ANL+. The WT ANL+ control for Figure 2F (previously 2G) is in the new Figure 2—figure supplement 3. We also added the relevant image to the modified Figure 3, now including the WT ANL+ control. We did not add the quantification because the CFP channel (not existing in WT) was used for the segmentation. We used “lookup table fire”, because otherwise the image would look too dark. Quantification of the ANL- fluorescence can be found in Figure 2H. We added quantification of Figure 2A in the new Figure 2—figure supplement 1.

2) Related to the point above, were the ANL- samples in Figures 1 and 2 processed for click-dye labeling?

Yes.

Why are the images completely dark?

They are not completely dark. It looks dark because the background is low in the absence of ANL.

3) In Figure 2G, authors show colocalization of CFP and click-label in confocal stacks made from 4-6 planes spanning several cell layer thickness. Single optical slices need to be shown.

We added as a new figure. Please see Figure 2—figure supplement 4.

4) Authors report average of averages in Figures 2H and 2I. As such, the variability in signal amplitude seen in different neurons will likely be much higher than the spread shown. It is not clear how much variability is seen within one region, say habenula, in a single larva. It will also be better to show the data scatter for the bar plots shown in 2H and 2I. This is now standard practice.

We modified the figure to include the average intensities of all the individual neurons. Please see the modified Figure 2. We moved the previous graphs to the supplementary figure: Figure 2—figure supplement 5.

5) Why were the spinal cord and habenula alone chosen for the PTZ experiment?

They were chosen as two exemplar regions with clear morphology and different density of neurons as well as different roles.

6) Authors call data in Figure 4D and E "Protein Synthesis Level". This is too strong – the signal likely reflects NET incorporation of ANL deriving from both synthesis and degradation of proteins.

We thank the reviewer for the comment. This is corrected.

7) Supplementary Figure 3 does not seem to be referred anywhere in the text.

It is referred to in the original manuscript: “Supplementary Figures 1C-D, 3”. In order to make it clear we now changed to “(Figure 2—figure supplement 2C-D, Figure 2—figure supplement 8).” (The previous Supplementary Figure 3 is now called “Figure 2—figure supplement 8”).

8) There are several places where the figure panels and their description in legends or the Results section are mismatched.

We apologize for this. It was corrected.